# LEARNING FLEXIBLE LARGE MULTIMODAL MODELS WITH ARBITRARY MODALITY COMBINATIONS

## ABSTRACT

Multimodal Large Language Models (MLLMs) show strong potential for cross-modal understanding by integrating powerful language models with multimodal encoders. However, extending MLLMs to handle a diverse range of modalities introduces two critical and intertwined challenges: (1) the reliance on fully paired multimodal data, often scarce or costly to acquire across all modalities, and (2) the computational inefficiency from processing numerous modality tokens and requiring substantial model updates for each new modality. To address these challenges, we enable MLLMs to handle missing modalities by generating representations for absent inputs. Furthermore, recognizing that an increasing number of modalities leads to linearly scaling token counts and that lengthy generated sequences can hinder performance, we employ a dual-stage compression mechanism. It first reduces the number of tokens per modality and then condenses information from multiple modalities into a single, compact token sequence. This culminates in Flex-M$^3$, a novel MLLM framework designed for flexible and efficient learning across arbitrary combinations of modalities. Experiments across diverse multimodal benchmarks and backbones demonstrate that Flex-M$^3$ robustly handles varied modality inputs and scales efficiently. Notably, Flex-M$^3$ outperforms its counterpart trained on only full-modality data, with consistent improvements of {2.29%, 3.15%, 11.01%} on multimodal reasoning tasks {`NExT-QA`, `MUSIC-AVQA`, `SQA3D`}. Moreover, Flex-M$^3$ model demonstrates superior robustness during inference, even when a high proportion of modalities are missing from the input samples. Codes are provided in the supplement material.

## 1 INTRODUCTION

In recent years, Multimodal Large Language Models (MLLM) have become a popular paradigm in multimodal learning. MLLMs leverage the understanding and generative capabilities of pre-trained Large Language Models (Dubey et al., 2024; Achiam et al., 2023; Anil et al., 2023), enhancing them by integrating information from diverse perceptual inputs (*e.g.*, vision (Liu et al., 2024; Wang et al., 2024), speech (Zhang et al., 2023a; Chu et al., 2023), 3D (Xu et al., 2024), biomarker (Zhuo et al., 2024), and tabular information (Zheng et al., 2024)). Recent advancements are pushing towards omnipotent MLLMs which manage numerous modalities to tackle complex scenarios, *i.e.* automated planning (Wei et al., 2024; Wang et al., 2023a) and world simulation (Ge et al., 2024).

However, realizing the full potential of MLLMs is challenged by data acquisition and training efficiency. Firstly, acquiring fully paired multimodal datasets is arduous. This could be attributed to real-world constraints, such as in biomedical settings where measurement devices might destroy paired samples (Xi et al., 2024). Furthermore, collection costs vary drastically across modalities. For example, readily available image-text pairs are far more abundant than data for depth or thermal imaging (Zhu et al., 2024; Girdhar et al., 2023). Prior work has explored data synthesis, image translation (Bhat et al., 2023; Xu et al., 2023; Lee et al., 2023a), or meticulous training pipelines over disparate data resources (Han et al., 2024) to mitigate this. However, these methods often involve laborious data preparation and empirical tuning of training dynamics, limiting their generalizability.

A second critical challenge is the substantial computational cost associated with training and deploying MLLMs. Incorporating each new modality requires significant updates to the LLM to align textual representations with the new modal input. While research into efficient MLLMs proposes using

separate projections or adapters to reduce trainable parameters (Li et al., 2023; Han et al., 2024; Yu et al., 2025), the inherent MLLM architecture that projects each modality into hundreds of tokens still leads to high training and inference costs. This is especially problematic with a growing number of modalities or computationally intensive modalities like video. Moreover, many efficient MLLMs lack flexibility, mandating the presence of all designated modalities, which restricts their use with a mixture of incomplete data.

In light of the above challenges, we posit that one critical next step for MLLMs reflecting real-world data scenarios is **"flexible multimodal learning"**, which is *enabling MLLMs to adeptly process diverse input samples, where each sample can present a different and potentially incomplete combination of available modalities.* To realize flexible multimodal learning, we introduce Flex-M$^3$ with a generation module synthesizing representations for any missing modalities by dynamically conditioning on the ones that are present. Then, we observed that the number of tokens, particularly those generated for missing inputs, significantly impacts training efficiency and final performance. As more modalities are introduced, this can lead to a linear scaling of tokens, and generating lengthy sequences for absent modalities can constrain performance. To mitigate this, Flex-M$^3$ incorporates a two-stage compression process. Initially, we compress the token representations from each modal encoder. Following that, all available modal representa-

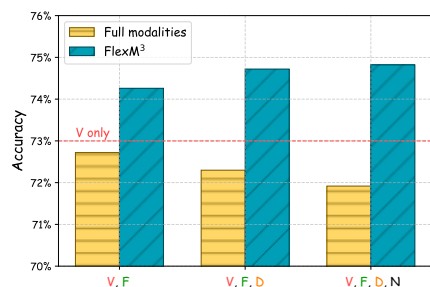

Figure 1: Comparison of accuracy (%) on a multimodal video question answering task NeXT-QA using different modality combinations for Flex-M$^3$ against a baseline trained on full modalities data only. The x-axis represents the available non-text modalities during fine-tuning: **V**ideo, optical **F**low, **D**epth, and surface **N**ormalization. The dashed red line indicates the performance when using only **V**.

tions, both those originally present and those newly generated, are further consolidated into a single, highly compact token sequence. This ensures that only the most salient and efficiently encoded cross-modal information is passed to the LLM. We validate the efficacy of Flex-M$^3$ across various MLLM backbones and diverse multimodal tasks. As illustrated in Figure 1, our approach not only robustly handles incomplete data but also achieves an average performance gain of nearly 3% compared to counterparts trained exclusively using full modality samples. This advantage becomes even more distinct in groups involving more modalities. In sum, the contributions of this study are four-fold:

• We formulate and advance flexible multimodal learning as a significant capability for MLLMs, empowering them to learn on data samples with diverse modality combinations, akin to real-world incomplete data scenarios.

• We develop Flex-M$^3$, a MLLM architecture to manage arbitrary combinations of input modalities. This is realized through a lightweight generation module that utilizes prompt-tuning to dynamically synthesize latent representations for absent modalities during both training and inference, effectively addressing the challenge of data scarcity.

• We introduce a two-stage token compression strategy integrated within Flex-M$^3$. It first condenses outputs from individual modality encoders, and then consolidates all present and generated modal information into a highly compact representation for the LLM, thereby enhancing computational efficiency and synthesis robustness.

• We conduct comprehensive empirical evaluations across several challenging multimodal VQA benchmarks. Employing different MLLM architectures (BLIP-2, LLaVA), Flex-M$^3$ consistently demonstrates superior performance and notable computational savings compared to baseline models restricted to full-modality training, achieving significant improvements in complex reasoning tasks (*e.g.*, an uplift of up to 11% on SQA3D).

## 2 RELATED WORK

**Multimodal Large Language Models**   Recent advancements in Multimodal Large Language Models (MLLMs) have streamlined the integration of diverse modalities, leading to improved performance in multimodal reasoning. BLIP-2 (Li et al., 2023) employs Querying Transformer (Q-Former) to bridge frozen image encoders and large language models. This design achieves competitive performance on vision-language tasks while maintaining a low number of trainable parameters. LLaVA (Liu et al., 2024) enables multimodal understanding by aligning image features

with language representations through a learned projection layer. This simple approach allows pre-trained language models to process image-text pairs effectively for VQA tasks. To process more modalities than vision and text, CREMA (Yu et al., 2025) proposes a modular and efficient framework. By employing modality-specific adapters on Q-Former and a multimodal fusion layer, CREMA can flexibly incorporate additional modalities—depth, flow, surface normal, audio, and 3D point clouds—without necessitating extensive parameter updates. However, all these models require the presence of all modalities during training and inference.

**Multimodal Learning with Missing Modalities** Real-world multimodal systems frequently face missing modalities due to factors such as environmental interference, sensor failures, or privacy constraints, all of which can significantly degrade model performance. Consequently, developing robust MLLMs capable of handling incomplete modality inputs has become a key research focus (Ma et al., 2022; Wei et al., 2023; Lee et al., 2023b; Qiu et al., 2023; Zhang et al., 2023c; Wu et al., 2024). Common recovery strategies include zero-based (Parthasarathy & Sundaram, 2020), average-based (Zhang et al., 2020), and learning-based methods (Pham et al., 2019). Among these, learning-based approaches are more effective, as they leverage representation learning and generative models to capture complex cross-modal dependencies. These methods can be broadly categorized into data-level and representation-level generation. Data-level methods aim to reconstruct the missing raw modalities from the available ones (Tran et al., 2017; Pham et al., 2019; Wang et al., 2023b), while representation-level methods synthesize the latent representations of missing modalities either directly from observed data (Hoffman et al., 2016; Zheng et al., 2021) or by fusing available modality representations (Zhou et al., 2021; Zhi et al., 2024). Recent works have explored architectural flexibility. Flex-MoE (Yun et al., 2024) utilizes a Mixture-of-Experts framework for medical classification with flexible modality inputs. PathWeave (Yu et al., 2024) enables models to continually evolve to incorporate new modalities. Our work targets complex MLLM reasoning tasks like video question answering and investigate robustly handling arbitrary combination of modalities.

**Efficient Multimodal Large Language Models** To support computation-heavy applications such as video understanding, recent work on MLLMs has focused on improving efficiency by reducing memory usage during training and inference. For image-based models, various techniques aim to reduce the number of vision tokens without sacrificing performance. Token pruning methods like FastV (Chen et al., 2024) discard less informative vision tokens in later attention layers, while token merging methods such as PruMerge (Shang et al., 2024) adaptively combine redundant tokens. TokenPacker (Li et al., 2024b) further compresses tokens through a coarse-to-fine approach. Other models, including Qwen-VL (Bai et al., 2023) and MQT-LLaVA (Hu et al., 2024), use Q-Former (Li et al., 2023) to project vision tokens into a fixed-length embedding. For video-based MLLMs, the challenge of processing long sequences of frames is addressed by selecting a fixed number of frames, as done in Video-ChatGPT (Maaz et al., 2024), VideoChat (Li et al., 2024a), Video-LLaVA (Lin et al., 2023), and Video-LLaMA (Zhang et al., 2023b), or by compressing the entire video into a compact representation, as in MovieChat (Song et al., 2024). LLaVA-Mini (Zhang et al., 2025) introduces a distinct strategy fusing visual information into text tokens and applying a query-based compression, reducing vision inputs to one token. This design enables highly compact multimodal representations and can potentially reduce the complexity of missing modality generation.

## 3 METHODOLOGY

We first provide a preliminary of the MLLM framework for connecting multimodal inputs with the LLM in Section 3.1. Then, we introduce Flex-M$^3$, starting with how to learn on arbitrary modality combination by generating missing modal embeddings in Section 3.2, followed by how to further enhance generation robustness using two stages compression in Section 3.3.

### 3.1 PRELIMINARY: MULTIMODAL LARGE LANGUAGE MODEL

Multimodal Large Language Models (MLLMs) extend LLMs to process and reason over multiple modalities such as vision, speech, and 3D data. Their architecture typically consists of modality-specific encoders, an interfacing module, and the LLM. Each encoder $\mathcal{E}_m$ maps raw inputs $\mathbf{X}_m$ into high-level features $\mathbf{F}_m = \mathcal{E}_m(\mathbf{X}_m)$. Pretrained on large unimodal or text-paired datasets, these encoders learn robust and meaningful feature representations for their respective modalities. The interfacing module $\mathcal{A}$ aligns these multimodal features with the LLM's input space by projecting them into token sequences or embeddings. This step may involve simple MLP projection layers or more sophisticated adaptors using modality-specific learnable queries $\mathbf{Q}_m$ that distill salient information

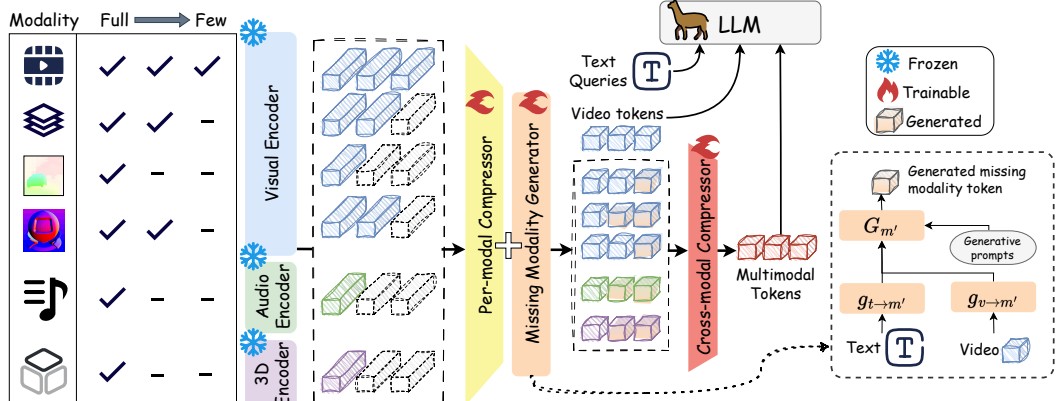

Figure 2: Overview of Flex-M³ multimodal learning framework. The model processes arbitrary modality combinations by first generating missing modality embeddings using text and video-conditioned generative soft prompts (left subfigure). These, along with present modality tokens, undergo per-modal and then cross-modal compression to create a compact, robust representation. Finally, these compressed tokens, along with text and video, are input to an LLM.

into fixed-length embeddings. The resulting tokens, $\mathbf{H}_m = \mathcal{A}(\mathbf{F}_m, \mathbf{Q}_m)$, act as soft prompts (Li et al., 2023) that condition the LLM on multimodal context.

These tokens $\mathbf{H}_m$ are then combined—typically concatenated or interleaved—with text embeddings $\mathbf{H}_t$ and fed into the LLM, which generates responses $\mathbf{Y}$. MLLMs are trained end-to-end using a language modeling objective: $\mathcal{L}_{\mathrm{LM}} = -\sum_{t=1}^{T} \log P(y_t \mid y_{<t}, \mathbf{H}_m, \mathbf{H}_t; \theta)$, enabling joint reasoning over text and other multimodal inputs.

## 3.2 LEARNING ON RANDOM MODALITY COMBINATION WITH MISSING MODALITY GENERATION

To address the challenge of incomplete data modalities, where one or more modalities may be absent, we introduce a generation module to recover representations for missing modalities from presenting ones. This approach allows the MLLM to effectively learn and operate across arbitrary combinations of available input data, significantly enhancing its flexibility.

The core of this generation process utilizing a set of consistently available modalities, *i.e.* text and video inputs as conditional information to recver other modalities as "supportive" modalities. For each target "extra" modality $m'$ (*e.g.*, depth, thermal, or other sensory data) that might be missing, we generate its feature representation. This generation is implemented by three components:

• **A learnable generative prompt** $\mathbf{P}$, which is a sequence of PROMPT vectors that provides an initial template or inductive bias for the generation process.

• **Modality transformation networks**: For each potential target missing modality $m'$, a dedicated mapping functions, $g_{t \to m'}(\cdot)$ and $g_{v \to m'}(\cdot)$, transform the text embedding $\mathbf{H}_t$ and projected visual embedding $\hat{\mathbf{H}}_v$ into representations suitable for conditioning the generation of $\hat{\mathbf{H}}_{m'}$. These mapping functions are implemented as Multi-Layer Perceptrons (MLP).

• **Final generation network**: The representations from the modality transformation networks are concatenated with the generative prompt $\mathbf{P}$ and then processed by another MLP $G_{m'}(\cdot)$ to produce the final synthesized feature embedding $\hat{\mathbf{H}}_m$.

In sum, the generation process for a missing modality $m'$ can be formulated as Equation 1, where concat denotes the concatenation operation along the sequence dimension.

$$\hat{\mathbf{H}}_{m'} = G_{m'} \left( \texttt{concat} \left( \mathbf{P}, g_{t \to m'}(\mathbf{H}_t), g_{v \to m'}(\mathbf{H}_v) \right) \right) \tag{1}$$

This architecture allows for the generation of multiple missing modalities, using the same set of source modalities and the shared generative prompt, but with distinct and lightweight transformation procedures. The generation modules are trained end-to-end with the rest of the MLLM. To enable learning on generating high quality missing modal embeddings, we employ a reconstruction objective.

During training, for data samples where a modality $m$ is physically present, we stochastically treat it as "missing". In such cases, we obtain a generated feature $\hat{\mathbf{H}}_m$. Then, we compute a reconstruction loss, typically the Mean Squared Error (MSE), between the generated features $\hat{\mathbf{H}}_m$ and the presented real features $\mathbf{H}_m$. This loss is formulated as in Equation 2. The overall training objective for the MLLM is a combination of the standard language modeling loss and the weighted reconstruction losses: $\mathcal{L} = \mathcal{L}_{\text{LM}} + \lambda\mathcal{L}_{\text{Rec}}$, where $\lambda$ is the weighting factor for reconstruction loss.

$$\mathcal{L}_{\text{Rec}} = \frac{1}{n}\sum_{m=1}^{n}\frac{1}{D}\sum_{i=1}^{D}\parallel \hat{h}_m^i - h_m^i \parallel \tag{2}$$

### 3.3 Modality Token Compression for Robust Generation

While the integration of multiple modalities enriches the context, the direct concatenation of all modality tokens can lead to a prohibitively large number of input tokens for the LLM. This not only escalates computational cost but can also introduce noise or redundant information, potentially hampering the robustness of the synthesized outputs. To mitigate these issues, we employ a two-stage strategy for compressing and refining modality tokens before they are processed by the main LLM. This strategy involves per-modality token compression and cross-modal token compression.

**Per-Modal Compression**   The initial projected feature representations for each modality $m$, denoted as $\mathbf{H}_m$, and including any generated features $\hat{\mathbf{H}}_{m'}$ are often lengthy. To reduce this length, we apply a per-modality compression module, $\mathcal{C}_m$. It is designed to distill the most salient information from $\mathbf{H}_m$ into a more compact representation, $\mathbf{H}_m^{(c)}$. The compression module employs a set of $N_q$ learnable query embeddings, *e.g.* $q_m \in \mathbb{R}^{N_q \times d}$ for modality $m$, where $d$ is the dimension of query embedding and $N_q$ is significantly smaller than the original token length of $\mathbf{H}_m$. These queries interact with the input modality tokens through the cross-attention mechanism. For the missing modalities, we switch to generate the per-modal compressor output $\hat{\mathbf{H}}_{m'}^{(c)}$.

**Cross-Modal Compression**   After per-modality condensation, concatenating the resulting tokens with text embeddings still lead to a long input sequence for the LLM, especially when there are more modalities. To further condense the input and enable earlier cross-modal interactions, we introduce a cross-modal compression stage. This stage creates a more integrated and compact set of supportive multimodal tokens before interacting with the LLM. In this stage, all compressed modal tokens are concatenated and then processed by a cross-modal compression module $f(\cdot)$, generating a fused multimodal token with fixed length for any input modality numbers $M$ as:

$$\mathbf{Z} = f(\texttt{concat}(\mathbf{H}_0^{(c)}, \mathbf{H}_1^{(c)}, \dots, \mathbf{H}_M^{(c)})) \tag{3}$$

The cross-modal compression output $\mathbf{Z}$, along with visual and text embeddings, are finally presented to the main LLM. This two-stage compression approach not only reduces the computational burden on the LLM but also aims to improve the robustness of generation by enabling the model to focus on the most salient cross-modal information, effectively filtering redundancies and noise.

## 4 Experiment

### 4.1 Experiment Setup

**Datasets Details**   We evaluate Flex-M$^3$ on the 3 multimodal video reasoning and QA tasks. Following the setup in Yu et al. (2025), we incorporate optical flow, depth maps, and surface normals extracted from the videos as additional modalities to enhance the model's understanding. Specifically, ZoeDepth (Bhat et al., 2023), Unimatch (Xu et al., 2023), and NLL-AngMF (Bae et al., 2021) are employed to extract depth, flow, and normal modalities.

• `NExT-QA` (Xiao et al., 2021) is a video question answering benchmark designed to advance video understanding beyond simple descriptions towards explaining temporal actions. It focuses on causal and temporal action reasoning as well as common scene comprehension. The dataset comprises 5440 videos and approximately 52K questions. We report the results on the validation set of `NExT-QA`.

• `SQA3D` (Ma et al., 2023) is a compositional VideoQA task centered around situated question answering within 3D scenes. It is built upon 650 scenes from ScanNet, featuring approximately 33K diverse reasoning questions, spanning a range of capabilities, including spatial relation comprehension, commonsense understanding, navigation, and multi-hop reasoning. Following (Hong et al., 2023b),

we utilize the ego-centric videos corresponding to the 3D scenes as video inputs. We report the results on the validation set.

• **MUSIC-AVQA** (li et al., 2022) is a compositional Audio-Visual Question Answering benchmark designed for comprehensive multimodal understanding and spatio-temporal reasoning over audio-visual scenes. It contains over 45K question-answer pairs derived from 9K videos. We train and evaluate baseline models and Flex-M$^3$ on the real video portion.

**Model Implementation and Training setup.** For **pretrained modal encoder**, we utilize ViT-G (Sun et al., 2023) for all visual modalities including videos, depth, norm and flow. For non-visual modalities, we use BEATs (Chen et al., 2023) as the encoder for audio, and extract 3D point cloud features offline following 3D-LLM (Hong et al., 2023a). For **MLLM model backbone**, we implement Flex-M$^3$ on BLIP-2 (Li et al., 2023) and LLaVA (Liu et al., 2024) to showcase Flex-M$^3$ general applicability, detailed training hyperparameters are shown in Table **??**. The entire model is trained end-to-end with the standard language modeling loss and an auxiliary generation reconstruction loss with weight $\lambda = 0.001$.

• We adapt BLIP-2's initial Q-Former architecture as per-modal compressor, with query token number $N_q = 32$. The cross-modal compressor is implemented as a modality-specific linear layer that projects the output features from the corresponding Q-Former into the language model. For fine-tuning, we initialize Flex-M$^3$ from BLIP-2 with encoders and LLM are frozen and only the per-modal compressors, cross-modal compressor, and generator are updated. To further enhance fine-tuning efficiency, we update per-modal compressors using LoRA (Hu et al., 2022) with rank 64.

• For LLaVA-based Flex-M$^3$, we similarly integrate our generation and compression mechanisms with Llama 3.1 8B language model and ViT-L visual processing pipeline, following (Liu et al., 2024). Similar to the settings in BLIP-2, we initiate LLaVA with pretrained per-modal and cross-modal compressor from LLaVA-Mini (Zhang et al., 2025), where we copy the compressors for modalities other than video. The per-modal compressor is a 2D perceiver-resampler network with $8 \times 8$ learnable queries as input, while the cross-modal compression module is a 4-layer Transformer decoder. We finetune Flex-M$^3$ for all model components on LLaVA except for the encoders, as we find the performance gain after enabling the language model to be updated is significant while the computation cost growth is moderate.

**Compared Baselines and Evaluation Setup** To support the effectiveness of Flex-M$^3$, we consider three groups of comparison baselines: (1) **Essential Modalities Only**: These models utilize the full dataset but are restricted to processing only the essential text and video modalities. This baseline is also evaluated on text and video modalities only. (2) **Full Modalities with Incomplete Data**: For $M$ available modalities, we simulate data scarcity across modality combinations. A standard Multimodal Large Language Model (MLLM), pre-trained on data with all $M$ modalities present, is subsequently fine-tuned. For this fine-tuning, the original dataset is divided into $2^M$ subsets, each corresponding to one of the $2^M$ possible modality input combinations and containing $1/2^M$ of the original data volume. For example, with one additional modality (e.g. Video, Flow in the first experiment group of NExT-QA) in Table 1, the data composition is 50% of the samples except texts contain V only while the rest of them contain all modalities. This baseline is evaluated on full modalities. (3) **Learnable Padding for Missing Modalities**: This baseline employs the full dataset while accommodating arbitrary modality combinations through a learnable padding technique. Specifically, [PAD] tokens from the LLM embedding space, are used to represent absent modalities. These padded inputs are then processed by the cross-modal compressor, enabling fusion of the padding with existing modalities. This improved baseline and Flex-M$^3$ are evaluated on full modalities.

### 4.2 MAIN RESULTS

**Superior Performance of Flex-M$^3$ Learned from Random Combinations of Modality Data** The fine-tuning results on NExT-QA, presented in Table 1, compellingly demonstrate that Flex-M$^3$ excels in handling various multimodal inputs, particularly in scenarios characterized by missing modalities. Taking Flex-M$^3$ with BLIP-2 as example, ❶ when utilizing the full dataset with arbitrary modality combinations (indicated by "Missing: ✓"), Flex-M$^3$ consistently outperforms alternative approaches. For instance, in the V, F, D, N setting, Flex-M$^3$ achieves an average score of 74.82, surpassing both the "Padding" baseline (74.22) and the "Essential Modalities Only" baseline (V: Avg. 73.00). This highlights Flex-M$^3$'s proficiency in leveraging supportive information from additional modalities, even when their presence is not guaranteed. ❷ This contrasts sharply with the "Full Modalities with Incomplete Data" baseline (rows with "Missing: ✗" and "Method: -"), which exhibits

Table 1: Performance on Video Question Answering (`NExT-QA`). Notations for each modality and question type are: **V**: Video RGB frames, **F**: optical Flow, **D**: Depth, and **N**: surface Normalization. **P.&N.**: Prev & Next, **Pre.**: Present, **Cnt.**: Count, **Loc.**: Location, and **Otr.**: Other. Within each experimental group, the best performance is indicated in **bold**, and the second-best result is underlined. All results are reported as percentages.

| Modality | Missing | Method | Causal | | | Temporal | | | Descriptive | | | | Avg. |
|---|---|---|---|---|---|---|---|---|---|---|---|---|---|
| | | | How | Why | Avg. | P.&N. | Pre. | Avg. | Cnt. | Loc. | Otr. | Avg. | |
| **BLIP-2** | | | | | | | | | | | | | |
| V | ✗ | - | 69.69 | 74.64 | 73.34 | 65.14 | 72.55 | 74.84 | 64.41 | 92.20 | 81.31 | 81.60 | 73.00 |
| V, F | ✗ | - | 69.55 | 74.43 | 73.15 | 64.58 | 72.85 | 74.20 | 66.10 | 91.53 | 79.67 | 81.08 | 72.72 |
| | ✓ | Padding | **71.16** | 74.58 | 73.69 | 65.25 | 73.00 | 74.97 | 64.97 | **92.54** | 81.64 | 81.98 | 73.26 |
| | ✓ | Flex-M³ | 70.42 | **75.99** | **74.53** | **66.93** | **73.60** | **76.89** | **65.54** | 92.20 | **83.28** | **82.63** | **74.26** |
| V, F, D | ✗ | - | 66.91 | 73.86 | 72.04 | 64.25 | 72.85 | 73.81 | **64.97** | 93.56 | 80.66 | 81.98 | 72.30 |
| | ✓ | Padding | 70.28 | **76.20** | 74.65 | 66.03 | **73.45** | 75.87 | 64.41 | **93.90** | 81.64 | 82.37 | 74.02 |
| | ✓ | Flex-M³ | **73.06** | 75.68 | **74.99** | **67.15** | 74.36 | **77.15** | 63.84 | 91.53 | **83.61** | 82.11 | **74.72** |
| V, F, D, N | ✗ | - | 66.91 | 73.86 | 72.04 | 64.25 | 72.85 | 73.81 | **64.97** | 93.56 | 80.66 | 81.98 | 72.30 |
| | ✓ | Padding | **72.62** | 75.31 | 74.61 | 66.59 | 74.51 | 76.51 | 63.28 | 93.22 | 81.97 | 81.98 | 74.22 |
| | ✓ | Flex-M³ | 70.66 | **76.62** | **75.06** | **66.85** | **74.81** | **76.84** | 64.41 | **93.90** | **84.92** | **83.66** | **74.82** |
| **LLaVA** | | | | | | | | | | | | | |
| V | ✗ | - | 74.38 | 77.23 | 76.49 | 68.60 | 75.17 | 71.53 | 59.32 | 93.22 | 84.92 | 82.24 | 75.78 |
| V, F | ✗ | - | 71.89 | 76.30 | 75.14 | 68.16 | 74.34 | 70.91 | 58.76 | 92.88 | 85.57 | 82.24 | 74.88 |
| | ✓ | Padding | 76.28 | 77.39 | 77.10 | **69.39** | 75.45 | **72.08** | 61.58 | 92.54 | 86.23 | 83.01 | 76.40 |
| | ✓ | Flex-M³ | **77.01** | **77.96** | **77.71** | 68.04 | **75.87** | 71.53 | **62.15** | **93.22** | 86.23 | **83.40** | **76.60** |
| V, F, D | ✗ | - | 64.71 | 69.13 | 67.97 | 61.45 | 66.53 | 63.71 | 55.37 | 88.81 | 78.36 | 77.09 | 68.01 |
| | ✓ | Padding | 75.7 | 77.34 | 76.91 | 70.39 | **77.55** | **73.57** | 61.58 | 92.54 | 86.23 | 83.01 | 76.78 |
| | ✓ | Flex-M³ | **77.89** | **78.33** | **78.21** | **71.28** | 75.17 | 73.01 | 60.45 | **92.88** | 86.23 | 82.88 | **77.26** |
| V, F, D, N | ✗ | - | 49.63 | 54.05 | 52.90 | 49.83 | 55.23 | 52.23 | 51.98 | 80.68 | 64.26 | 67.70 | 54.98 |
| | ✓ | Padding | 74.38 | **78.22** | 77.22 | 68.27 | 76.15 | 71.77 | 62.71 | 92.2 | **88.85** | **84.17** | 76.56 |
| | ✓ | Flex-M³ | **77.89** | 77.91 | **77.91** | **70.84** | **76.43** | **73.33** | **63.84** | **92.54** | 86.56 | 83.66 | **77.32** |

a performance decline as more modalities are introduced (from 73.00 for V only, down to 72.04 for V, F, D, N). This performance drop could be attributable to the MLLM being fine-tuned on progressively smaller, specific data subsets for each modality combination ($1/2^M$ of the original data volume), which hampers generalization. ❸ Flex-M³ not only overcomes this limitation but also consistently betters the "Padding" method across all tested auxiliary modality counts: achieving a +1.00 point gain with one auxiliary modality (V, F: Flex-M³ 74.26 vs. Padding 73.26) and a +0.60 point gain with three (V, F, D, N: Flex-M³ 74.82 vs. Padding 74.22). This sustained advantage is attributed to Flex-M³'s modal specific generation and compression design, which effectively distills key information and manages modality absence more adeptly than simple learnable padding. ❹ Furthermore, this robust performance extends across diverse question categories (Causal, Temporal, Descriptive Average Performance), where Flex-M³ generally secures the highest scores in settings with multiple potential modalities. In essence, Flex-M³ showcases a significant capability in flexibly and efficiently integrating information from an arbitrary set of available modalities, underscoring the efficacy of its advanced modality compression techniques for robust multimodal understanding in the face of incomplete data.

**Generalization of Flex-M³ across Different MLLM Backbones** To further substantiate the generalizability of Flex-M³, we evaluated its efficacy when integrated with LLaVA architecture (Liu et al., 2024). The results presented in Table 1 (bottom), again validate the effectiveness of Flex-M³ against strong video-LLMs fine-tuned with extra supportive modalities. Flex-M³ with LLaVA demonstrate a substantial average performance increase of approximately 10.83% points compare to training with full modality samples only. This consistent improvement demonstrates that the architectural benefits of Flex-M³ can be effectively transferred across foundational models.

**Generalization of Flex-M³ across Non-visual modalities** To further evaluate whether Flex-M³ could extend to non-visual modalities that the model backbone has not been pre-trained on fine-tuned on, we perform fine-tuning and evaluation the `MUSIC-AVQA` and `SQA3D` benchmarks. Experiment results have been listed in Table 2 and Table 3. ❶ On the `MUSIC-AVQA` benchmark, Flex-M³ demonstrates its surprising capacity for audio-video reasoning. When leveraging auxiliary modality information where samples contain missing modalities, Flex-M³ achieve over 11% the baseline learned on full-modality data only. Also, compared to the baseline finetuned on text-video modality,

Table 2: Performance on Audio-Video Question Answering (MUSIC-AVQA) with BLIP-2-based Flex-M³ and baseline models. Notations for each modality and question type are: **V**: *Video RGB frames*, **A**: *Audio*, **F**: *optical Flow*, **D**: *Depth*, and **N**: *surface Normalization*. **Cnt.**: *Counting*, **Com.**: *Comparative*, **Loc.**: *Location*, **Ext.**: *Existential*, and **Tem.**: *Temporal*. Within each experimental group, the best performance is indicated in **bold**, and the second-best result is underlined. All results are reported as percentages (%).

| Modality | Missing | Method | Audio | | | Visual | | | Audio-Visual | | | | | | Avg. |
|---|---|---|---|---|---|---|---|---|---|---|---|---|---|---|---|
| | | | Cnt. | Com. | Avg. | Cnt. | Loc. | Avg. | Cnt. | Ext. | Loc. | Com. | Tem. | Avg. | |
| V | ✗ | - | 88.14 | 60.73 | 82.21 | 85.73 | 87.11 | 86.40 | 82.93 | 84.34 | 69.66 | 62.35 | 73.04 | 74.65 | 76.28 |
| V, A, F, D, N | ✗ | - | 79.75 | 57.09 | 74.85 | 75.75 | 77.05 | 76.38 | 69.65 | 80.54 | 58.71 | 56.38 | 68.18 | 66.79 | 70.93 |
| | ✓ | Padding | 89.49 | **65.18** | **84.22** | **87.03** | 90.43 | 88.69 | **85.67** | 83.11 | 71.49 | **67.59** | 73.04 | 76.54 | 81.17 |
| | ✓ | Flex-M³ | **89.71** | 62.75 | 83.87 | **87.03** | 92.48 | 89.69 | 84.93 | 85.12 | 73.74 | 66.87 | 74.14 | 77.19 | 81.94 |

Table 3: Performance on Situated Question Answering (SQA3D) with BLIP-2-based Flex-M³ and baseline models. Notations for each modality and question type are: *Video RGB frames*, **V**: *Bird-Eye View image*, **P**: *3D Point cloud*, **D**: *Depth*, and **N**: *surface Normalization*. Within each experimental group, the best performance is indicated in **bold**, and the second-best result is underlined. All results are reported as percentages (%).

| Modality | Missing | Method | What | Is | How | Can | Which | Others | Avg. |
|---|---|---|---|---|---|---|---|---|---|
| V | ✗ | - | 44.99 | 47.74 | 63.02 | 64.88 | 47.59 | 49.11 | 51.69 |
| V, P, D, N | ✗ | - | 43.59 | 45.38 | 63.02 | 59.97 | 50.42 | 49.29 | 50.33 |
| | ✓ | Padding | 45.86 | 45.81 | 65.98 | **65.95** | **49.86** | **54.26** | 53.25 |
| | ✓ | Flex-M³ | **46.82** | **47.74** | **66.57** | **65.95** | 47.03 | 53.55 | **53.48** |

Flex-M³ obtain performance gain comprehensively in all question subclasses (audio, visual, audio-visual). This again validates the benefit of utilizing mixture of modality combination data, and the potential of flexible multimodal learning. ❷ The results in SQA3D again validate the versatility and effectiveness of Flex-M³, where it achieves the leading average accuracy of 53.48% (+3.15% to full-modal data baseline). 3D-associated video reasoning tasks require a model to interpret dynamic visual narratives from video with static and rich spatial, geometric information from 3D modalities. The ability of Flex-M³ to leverage these combined inputs allows it to construct a more holistic and nuanced understanding of the scene. Computation analysis of Flex-M³ is provided in Appendix A.1.

**Flex-M³ Demonstrates Superior Robustness to Missing Modalities During Inference**
While from the evaluation with full modalities in Table 1-3, both padding and Flex-M³ outperform other baselines, the distinction emerges when assessing their performance under random modality absence during inference. We take NExT-QA with V, F, D, N modalities as example. We randomize missing conditions for each sample, where 1 to 3 supportive modalities (from F, D, N) could be absent. We use the Missing Ratio (MR) to denote the overall proportion of missing modalities across the entire test set. As depicted in Figure 3, the performance of naive padding approaches degrades as the MR

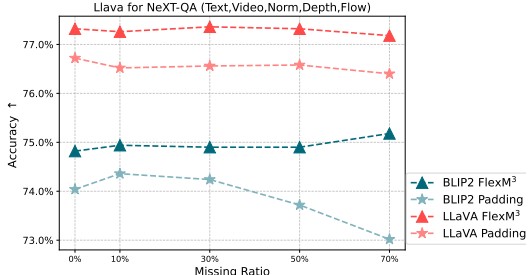

Figure 3: Comparison between BLIP-2 and LLaVA-based Flex-M³ and Padding baseline on random modality missing evaluation.

increases, a trend observed across both LLaVA and BLIP-2 based models. In contrast, Flex-M³, leveraging modal-specific generation, exhibits robust performance in both settings. The accuracy of Flex-M³ models remains stable or even slightly increases under high MR situations (70%), consistently outperforming the padding counterparts. This underscores a key advantage of Flex-M³. While naive padding falters with substantial data incompleteness at inference, Flex-M³ can manage modality variations through generation, providing a more resilient framework for MML.

### 4.3 EXTRA ANALYSIS AND ABLATION STUDIES

To identify the optimal design of Flex-M³, we analyze its module contributions, hyperparameter sensitivity, and training efficiency. All experiments are conducted with the BLIP-2 backbone on the 10% NExT-QA subset, trained for 5 epochs using all supportive modalities.

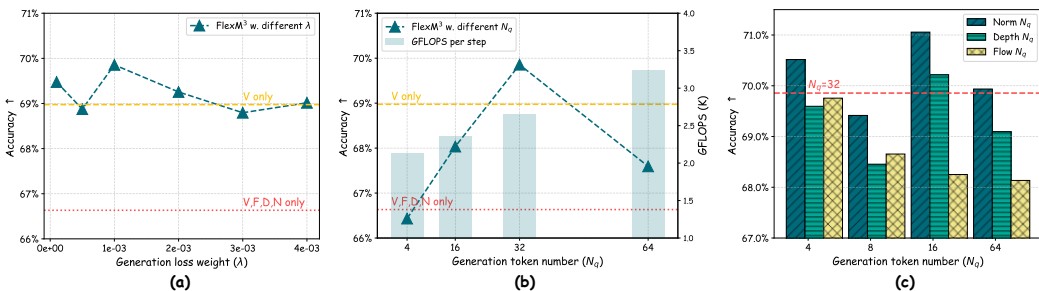

Figure 4: Extra studies on Flex-M$^3$ hyperparameters. **(a)** investigates the effect of varying the generation loss weight ($\lambda$) on model performance. **(b)** examines the impact of different generation token numbers ($N_g$) on accuracy and computational cost (GFLOPS per training step). **(c)** compares the impact of separately changing generation tokens per modality while keeping other modalities $N_q$=32. All experiments are conducted on NeXT-QA with V, F, D, N modalities.

**Ablation on Flex-M$^3$ Components.** We investigate the contributions of all modules within Flex-M$^3$, with results in Table 4. We begin by directly generating multimodal encoder outputs and concatenating them as inputs for the Large Language Model (LLM). Subsequently, we integrate a per-modal compressor while keeping the concatenation, followed by incorporating a cross-modal compression mechanism. The experimental results demonstrate that the synergistic combination of these design elements achieves a Pareto-optimal balance between computational efficiency and model performance.

Table 4: Ablation of model modules.

| Method | Avg. (%) |
|---|---|
| Baseline w/ full data | 68.98 |
| Baseline w/ full modalities | 66.63 |
| + Generation | 68.21 |
| + Per-modal compression | 68.43 |
| + Cross-modal compression | **69.86** |

**Ablation on Generation Loss Weight.** An appropriate choice of the generation loss weight could benefit the performance of Flex-M$^3$. We compare Flex-M$^3$ under different generation loss weights ($\lambda$) in Figure 4 (a). The results indicate that a moderate weight ($1e^{-3}$) appears to yield optimal accuracy. Performance drops noticeably when $\lambda$ is either significantly lower or higher. This suggests that while the reconstruction loss is crucial for learning to recover missing modalities, its contribution must be carefully balanced against the primary language modeling objective to prevent it from interfering with the core task.

**Generation token numbers.** In Figure 4 (b), we study the impact of the number of generation tokens ($N_q$) for all modalities on both accuracy and computational cost. As the token number increases to 32, accuracy generally improves. However, further increasing $N_q$ to 64 results in a slight decrease in accuracy. This suggests that $N_q = 32$ reaches an optimal balance between representational capacity for the generated tokens and computational efficiency, with larger values potentially introducing redundancy. Moreover, we investigate altering the generation token numbers across modalities while keeping the token numbers for other modalities fixed at 32. The results in Figure 4 (c) highlight how individual modalities could benefit from different representational capacities during generation. Overall, $N_q = 32$ achieves moderately high accuracy for all modalities, and more tokens do not guarantee higher performance, aligning with previous findings in Figure 4 (b). Interestingly, some modalities could even improve with smaller $N_q$. For example, $N_q = 16$ yields better results for the Norm and Depth modalities. These findings suggest that we could dynamically adjust the generation token number per modality for flexible multimodal learning.

## 5 CONCLUSION

Existing multimodal MLLMs necessitate complete sets of modal inputs for training and inference, limiting their ability to utilize the prevalent heterogeneous and incomplete multimodal data. This paper introduced Flex-M$^3$, a novel MLLM designed to adeptly process data featuring arbitrary combinations of modalities. Extensive experiments demonstrate that Flex-M$^3$ achieves significant performance gains across various MLLM backbones and diverse multimodal benchmarks, all while incurring minimal additional computational overhead.

## REPRODUCIBILITY STATEMENT

We have made efforts to ensure the methods and results in this paper are reproducible. Section 4.1 provides extensive details about the datasets (NEXT-QA, SQA3D, MUSIC-AVQA), including the specific tools used for preprocessing auxiliary modalities like optical flow and depth maps. The same section also guides readers through the model implementation, training setup, and evaluation procedures for the BLIP-2 and LLaVA backbones. The core architectural components of Flex-M$^3$ are detailed in Section 3. To facilitate replication, the source code to train and evaluate our models is included in the supplementary materials.

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

# A  APPENDIX

## A.1  EFFICENCT FLEXIBLE MULTIMODAL LEARNING WITH FLEX-M$^3$

To investigate the performance-computation trade-off of our generation framework, we list the parameters and the number of floating point operations (GFLOPs) per training forward of Flex-M$^3$ with two backbones in Table 5. From the results, we find out that both compression and generation methods (Padding and Flex-M$^3$) incur minimal computation overhead compared to its original architecture. Especially, for BLIP-2-based architectures, with LLM frozen and PEFT techniques, we could further improve training efficiency by updating less than $1\%$

Table 5: Comparison between Flex-M$^3$ and baselines on training cost on NeXT-QA with V, F, D, N modalities. $p_{total}$ refers to total parameters (M) and $p_{train.}$ indicates all trainable parameters (M).

| Modality | Avg. | $p_{total}$ | $p_{train.}$ | GFLOPs |
|---|---|---|---|---|
| BLIP-2 | 72.30 | 3947.65 | 16.83 | 2.47K |
| *w/* Padding | 74.22 | 3957.62 | 16.84 | 2.47K |
| *w/* Flex-M$^3$ | 74.82 | 3966.06 | 25.27 | 2.60K |
| LLaVA | 54.98 | 9307.47 | 9003.96 | 11.35K |
| *w/* Padding | 76.72 | 9307.47 | 9003.96 | 11.35K |
| *w/* Flex-M$^3$ | 77.04 | 9307.58 | 9004.07 | 11.35K |

parameters, while still benefiting from the multimodal learning performance gains.

## A.2  THE USAGE OF LLM

To enhance the clarity and readability of this manuscript, GPT-5 was utilized exclusively as a language polishing tool. Its role was strictly confined to proofreading, grammatical correction. GPT-5 did not contribute to the generation of any scientific content, experimental design, or new ideas presented in the paper. Its usage is consistent with standard practices for manuscript preparation and did not influence the research itself.

