# OpenReview forum: "Learning Flexible Large Multimodal Models with Arbitrary Modality Combinations"
_ICLR.cc/2026/Conference — ICLR 2026 Conference Withdrawn Submission_

### Official Review · Reviewer_L2Sb · 2025-10-23

**Soundness:** 2
**Presentation:** 2
**Contribution:** 2
**Rating:** 2
**Confidence:** 4

**Summary:**

The authors propose Flex-M3, a flexible and efficient Multimodal Large Language Model (MLLM) that can handle arbitrary combinations of modalities. To overcome the reliance on fully paired multimodal data and the inefficiency from growing token counts, Flex-M3 generates representations for missing modalities and applies a dual-stage compression mechanism to condense information. Experiments on NExT-QA, MUSIC-AVQA, and SQA3D show consistent gains of 2.29%, 3.15%, and 11.01%, respectively. The model also demonstrates strong robustness even when many modalities are missing.

**Strengths:**

1. Flexibility and robustness: Flex-M3 effectively handles missing modalities by generating substitute representations, enabling stable performance under incomplete multimodal inputs.

2. Efficiency and scalability: The dual-stage compression mechanism reduces token redundancy and computational cost, allowing efficient learning and inference as the number of modalities increases.

**Weaknesses:**

1. Limited performance improvement: Flex-M3 achieves less than 1% average gain over the “Padding” baseline across tasks, indicating modest practical enhancement despite the added complexity.

2. Outdated experimental setup: The model is built on older architectures, lacking evaluation on recent strong baselines such as Qwen2.5-VL, Qwen3-VL, and LLaVA-OV. Recent MLLMs can already flexibly handle missing modalities by inputting available ones only, which should be included as a fair and stronger baseline.

**Questions:**

I have no questions.

---

### Official Review · Reviewer_cdLh · 2025-10-25

**Soundness:** 3
**Presentation:** 3
**Contribution:** 3
**Rating:** 6
**Confidence:** 5

**Summary:**

This paper proposes Flex-M3, a flexible MLLM that processes arbitrary modality combinations by generating embeddings for missing inputs and compressing tokens in two stages. A generative module conditions on always-available text and video to synthesize latent features for absent modalities; per-modal and cross-modal compressors produce a compact sequence for the LLM. Evaluated on NExT-QA, MUSIC-AVQA, and SQA3D with BLIP-2 and LLaVA backbones, Flex-M3 improves accuracy over full-modality baselines and remains robust when many modalities are missing, with up to ~11% gains on SQA3D.

**Strengths:**

**1. Flexible missing-modality generation.** Flex-M3 synthesizes latent embeddings for absent modalities using learnable generative prompts conditioned on text and video, trained with an MSE reconstruction objective. This design lets the same backbone learn from arbitrary modality subsets without bespoke data pairing or model re-wiring.

**2. Two-stage compression for efficiency and robustness.** Per-modal compressors distill tokens with learned queries; a cross-modal compressor fuses all supportive modalities into a compact sequence fed to the LLM, reducing token counts and filtering redundancy while preserving salient cross-modal context.

**3. Consistent, backbone-agnostic gains.** Across BLIP-2 and LLaVA, Flex-M3 outperforms “essential modalities only,” “full-modalities with incomplete data,” and learnable-padding baselines on NExT-QA; it also improves MUSIC-AVQA and SQA3D and stays strong under high missing-ratio inference.

**4. Practicality and training cost awareness.** The method updates lightweight generators and compressors (with LoRA where applicable) and reports computational savings/parameter efficiency alongside accuracy, aiding reproducibility and adoption.

**Weaknesses:**

**1. Novelty of method.** The generator uses MLPs over text/video with a learnable prompt; compression uses query-based distillation. Compared with prior missing-modality and token-compression work, the advance reads as integrative rather than algorithmically novel.

**2. Comprehensiveness of the benchmark.** Evaluation focuses on three video-QA tasks (NExT-QA, MUSIC-AVQA, SQA3D) and auxiliary signals (flow, depth, normals). Broader domains—egocentric ASR, robotics, medical, documents, meetings—are absent, limiting claims about general flexible multimodality.

**3. Baseline construction and fairness.** The “full-modalities with incomplete data” baseline splits training into 2^M subsets per combination, likely handicapping it; stronger baselines (shared training with masking or curriculum over mixtures) would better isolate Flex-M3’s margin.

**4. Anchoring to text+video.** Generation assumes text+video are consistently available; effects when an anchor is absent—or when anchoring on non-visual modalities—remain unclear for audio-first or sensor-first deployments.

**Questions:**

**1. How does Flex-M3 behave if only text or only video is present, or when audio/3D serve as anchors? Can the generator flexibly switch anchors at inference without retraining?**

**2. What are end-to-end latency and accuracy trade-offs as query counts per modality and fused-token length vary on hour-long videos, and can adaptive token budgets per modality improve robustness under strict compute?**

**3. Will future releases include audio with ASR lag, robotics/control, meetings, and medical datasets to test missing-anchor cases, timestamp noise, and reliability across domains, and to compare against masking-based baselines trained on unified mixtures?**

---

### Official Review · Reviewer_atJs · 2025-10-28

**Soundness:** 2
**Presentation:** 3
**Contribution:** 2
**Rating:** 4
**Confidence:** 4

**Summary:**

This paper proposes Flex-M³, a framework designed to enable MLLMs to handle arbitrary modality combinations by addressing two key challenges: scarcity of fully paired multimodal data and computational inefficiency from growing modality tokens. To tackle data scarcity, Flex-M³ introduces a generation module that synthesizes representations for missing modalities using text and video (as fixed available modalities) as conditional inputs. For computational efficiency, it adopts a two-stage compression mechanism: first compressing tokens per modality via cross-attention, then consolidating all (original and generated) modal representations into a compact sequence. The authors evaluate Flex-M³ on three benchmarks (NExT-QA, MUSIC-AVQA, SQA3D) with BLIP-2 and LLaVA backbones, showing performance gains over baselines and robustness to missing modalities. Codes are provided in the supplement.

**Strengths:**

- The paper maintains a clear and logical structure, with well-organized sections (Introduction, Methodology, Experiments) that make it easy to follow the technical details and experimental design. Key concepts (e.g., two-stage compression, missing modality generation) are explained with sufficient clarity for readers to understand the framework’s workflow.
- The experimental setup is relatively detailed, including specific choices of datasets, backbones, and hyperparameters (e.g., reconstruction loss weight λ=0.001, per-modal compression query number N_q=32). This level of detail helps improve the transparency of the work, even if the baseline comparisons are limited.

**Weaknesses:**

- Lack of Innovation in Modality Compression: The two-stage compression mechanism relies on conventional cross-attention for per-modal compression and a linear layer/Transformer decoder for cross-modal compression. These designs are not novel—cross-attention-based token reduction (e.g., Q-Former in BLIP-2) and linear projection for feature fusion have been widely used in existing MLLM works (e.g., LLaVA-Mini, CREMA). The paper does not introduce any modified or adaptive compression logic to address the unique challenges of arbitrary modality combinations, making this component incremental at best.
- Rigidity in Conditional Modalities for Missing Modality Generation: The framework assumes text and video arefixed available modalities to generate missing modalities. However, real-world scenarios often involve missing of these "conditional modalities" (e.g., a sample with only audio and 3D point clouds but no video/text). The paper provides no solution for such cases, severely limiting Flex-M³’s practicality—its "flexibility" is constrained to scenarios where text and video are always present, which contradicts the goal of handling "arbitrary" modality combinations.
- Oversimplified and Redundant Missing Modality Generation: The generation of missing modalities uses a simple MLP-based pipeline with MSE reconstruction loss. This approach effectively replicates the function of task-specific models (e.g., ZoeDepth for depth estimation, Unimatch for optical flow), but the paper does not clarify: (1) Whether a separate MLP is trained for each target modality (e.g., one net for depth, another for flow) or a unified MLP for all modalities. (2) Why retraining these MLP-based generators is necessary, rather than leveraging pre-trained task-specific models (e.g., using ZoeDepth to directly extract depth instead of generating it via MLP). This redundancy raises questions about the framework’s efficiency and necessity.
- Inadequate Baseline Comparisons for Missing Modality Handling: The baselines for missing modalities are overly simple (e.g., "learnable padding with [PAD] tokens"). The paper fails to compare with state-of-the-art methods for missing modality learning in MLLMs, such as: (1)Representation-level generation methods (e.g., Borrowing Treasures from Neighbors’s in-context learning for missing modalities).(2)Architectural adaptations (e.g., Flex-MoE’s Mixture-of-Experts for flexible modality combinations). Without these comparisons, it is impossible to verify whether Flex-M³’s performance gains are due to its design or the weakness of the baselines.

**Questions:**

- Regarding the two-stage compression: Given that cross-attention and linear projection are conventional methods, how do you justify the novelty of your compression design? Do you have any ablation studies showing that your compression outperforms existing off-the-shelf compression modules (e.g., Q-Former in BLIP-2, the perceiver-resampler in LLaVA) when applied to arbitrary modality combinations?
- The missing modality generation relies on text and video as fixed conditionals. How would your framework handle cases where text or video is missing (e.g., a sample with only audio and 3D point clouds)? Do you have plans to extend the generator to adapt to dynamic conditional modalities (i.e., no fixed "core modalities")?
- For the MLP-based generator: a. Do you train a separate MLP for each target modality (e.g., depth, flow) or a unified MLP? Please clarify the model architecture and training pipeline for the generator. b. Why not use pre-trained task-specific models (e.g., ZoeDepth for depth, Unimatch for flow) to replace your MLP generator? Could you provide a comparison between your generated modalities and those extracted by pre-trained models, both in terms of quality and impact on downstream QA performance?
- The baselines for missing modalities are too simple. Could you supplement experiments comparing Flex-M³ with state-of-the-art missing modality learning methods  (e.g., Borrowing Treasures from Neighbors; Flex-MoE; PathWeave)? This is critical to validate the advantage of your approach over existing solutions.

[1] Borrowing Treasures from Neighbors: In-context Learning for Multimodal Learning with Missing Modalities and Data Scarcity (Zhi et al., 2024)

[2] Flex-MoE: Modeling Arbitrary Modality Combination via the Flexible Mixture-of-Experts (Yun et al., 2024)

[3] LLMs Can Evolve Continually on Modality for X-Modal Reasoning (Yu et al., 2024) (PathWeave)

[4] ZoeDepth: Zero-shot Transfer by Combining Relative and Metric Depth (Bhat et al., 2023)

[5] Unifying Flow, Stereo and Depth Estimation (Xu et al., 2023) (Unimatch)

[6] BLIP-2: Bootstrapping Language-Image Pre-training with Frozen Image Encoders and Large Language Models (Li et al., 2023)

[7] Improved Baselines with Visual Instruction Tuning (Liu et al., 2024) (LLaVA)

[8] LLaVA-Mini: Efficient Image and Video Large Multimodal Models with One Vision Token (Zhang et al., 2025)

[9] CREMA: Generalizable and Efficient Video-Language Reasoning via Multimodal Modular Fusion (Yu et al., 2025)

---

### Official Review · Reviewer_dA81 · 2025-10-31

**Soundness:** 2
**Presentation:** 3
**Contribution:** 2
**Rating:** 4
**Confidence:** 5

**Summary:**

This paper introduces Flex-M3, a framework for multimodal large language models (MLLMs) that can handle arbitrary combinations of input modalities, including cases with missing ones. The method generates latent representations for absent modalities through a lightweight generative prompt module and employs a two-stage compression strategy to reduce token redundancy across modalities. Experiments on multiple benchmarks (NExT-QA, MUSIC-AVQA, SQA3D) show consistent improvements over baselines, with better robustness when modalities are missing.

**Strengths:**

1. The paper is clearly written and well-organized.

2. The idea of flexible multimodal training that tolerates missing modalities is timely and relevant. The generative prompt mechanism for reconstructing missing modality embeddings is a reasonable extension of prior “hallucination” or imputation approaches.

3. The experimental evaluation is decent, spanning three benchmarks and two MLLM backbones (BLIP-2 and LLaVA). Implementation details are described with fair transparency.

4. The notion of “flexible multimodal learning” could be of interest to practitioners dealing with incomplete data; however, the improvement margins are modest and may not justify the architectural complexity.

**Weaknesses:**

1. The proposed generation and compression mechanisms combine existing ideas (prompt-based latent synthesis and token compression) rather than introducing a fundamentally new principle. Related works like Flex-MoE (Yun et al., 2024) or modality hallucination frameworks already tackle similar issues.

2. The empirical results, while numerous, mostly show small gains (~1–3%) over learnable-padding baselines. There is little ablation on how much each proposed component (generation vs. compression) contributes under real missing-modality conditions.

3. An experiment with recent top-tier MLLMs like Qwen2.5vl, Internvl3 is missing, which would better situate the performance gains within the current landscape.

4. Several implementation details (e.g., exact λ values, data splits for partial modality setups, or full hyperparameter tables) are missing or incomplete.

**Questions:**

1. Could the authors clarify how the generation module generalizes when all conditioning modalities differ from those seen during training?

2. How does Flex-M3 perform when non-visual modalities dominate (e.g., audio + text only)?

3. Please quantify the computational overhead of the generation and compression stages compared to a simple padding baseline.

4. Have the authors considered more recent top-tier baselines?

5. It would help to visualize qualitative examples of generated modality embeddings or attention maps to support interpretability claims.

6. Can the method handle dynamically varying numbers of modalities at inference without retraining the compressors?

---

### Note · Authors · 2026-01-06

I have read and agree with the venue's withdrawal policy on behalf of myself and my co-authors.